# Pulmonary Nodule and Mass: Superiority of MRI of Diffusion-Weighted Imaging and T2-Weighted Imaging to FDG-PET/CT

**DOI:** 10.3390/cancers13205166

**Published:** 2021-10-14

**Authors:** Katsuo Usuda, Masahito Ishikawa, Shun Iwai, Aika Yamagata, Yoshihito Iijima, Nozomu Motono, Munetaka Matoba, Mariko Doai, Keiya Hirata, Hidetaka Uramoto

**Affiliations:** 1Department of Thoracic Surgery, Kanazawa Medical University, Ishikawa 920-0293, Japan; masa-i@kanazawa-med.ac.jp (M.I.); mhg1214@kanazawa-med.ac.jp (S.I.); aicarby@kanazawa-med.ac.jp (A.Y.); y-iijima@kanazawa-med.ac.jp (Y.I.); motono@kanazawa-med.ac.jp (N.M.); hidetaka@kanazawa-med.ac.jp (H.U.); 2Shimada Hospital, Fukui 910-0855, Japan; 3Department of Radiology, Kanazawa Medical University, Ishikawa 920-0293, Japan; m-matoba@kanazawa-med.ac.jp (M.M.); doaimari@kanazawa-med.ac.jp (M.D.); 4MRI Center, Kanazawa Medical University Hospital, Ishikawa 920-0293, Japan; keiya@kanazawa-med.ac.jp

**Keywords:** positron emission tomography–computed tomography (FDG-PET/CT), magnetic resonance imaging (MRI), diffusion-weighted magnetic resonance imaging (DWI), T2-weighted imaging (T2WI), lung cancer, pulmonary nodule and mass (PNM)

## Abstract

**Simple Summary:**

Although diffusion-weighted imaging (DWI) can be valuable for differential diagnosis of lung cancer from benign pulmonary nodules and masses (PNMs), the diagnostic capability may not be perfect. This study’s purpose was to compare the diagnostic efficacy of 18-fluoro-2-deoxy-glucose positron emission tomography–computed tomography (FDG-PET/CT) and magnetic resonance imaging (MRI) of DWI and T2-weighted imaging (T2WI) in PNMs. There were 278 lung cancers and 50 benign PNMs that were examined by FDG-PET/CT and MRI. The sensitivity of the maximum standardized uptake value (SUVmax) was significantly lower than that of the apparent diffusion coefficient (ADC) and the T2 contrast ratio (T2 CR). The accuracy of SUVmax was significantly lower than that of ADC and that of T2 CR. The sensitivity and accuracy of MRI were significantly higher than those of FDG-PET/CT. MRI can replace FDG-PET/CT for differential diagnosis of PNMs.

**Abstract:**

The purpose of this retrospective study was to compare the diagnostic efficacy of FDG-PET/CT and MRI in discriminating malignant from benign pulmonary nodules and masses (PNMs). There were 278 lung cancers and 50 benign PNMs that were examined by FDG-PET/CT and MRI. The T2 contrast ratio (T2 CR) was designated as the ratio of T2 signal intensity of PNM divided by T2 signal intensity of the rhomboid muscle. The optimal cut-off values (OCVs) for differential diagnosis were 3.605 for maximum standardized uptake value (SUVmax), 1.459 × 10^−3^ mm^2^/s for apparent diffusion coefficient (ADC), and 2.46 for T2 CR. Areas under the receiver operating characteristics curves were 67.5% for SUVmax, 74.3% for ADC, and 72.4% for T2 CR, respectively. The sensitivity (0.658) of SUVmax was significantly lower than that (0.838) of ADC (*p* < 0.001) and that (0.871) of T2 CR (*p* < 0.001). The specificity (0.620) of SUVmax was that the same as (0.640) ADC and (0.640) of T2 CR. The accuracy (0.652) of SUVmax was significantly lower than that (0.808) of ADC (*p* < 0.001) and that (0.835) of T2 CR (*p* < 0.001). The sensitivity and accuracy of DWI and T2WI in MRI were significantly higher than those of FDG-PET/CT. Ultimately, MRI can replace FDG PET/CT for differential diagnosis of PNMs saving healthcare systems money while not sacrificing the quality of care.

## 1. Introduction

18-fluoro-2-deoxy-glucose positron emission tomography–computed tomography (FDG-PET/CT) is widely accepted as the imaging modality of choice in tumor staging and as differentiating between malignant and benign pulmonary nodules [1,2]. FDG-PET/CT has a high false-positive rate and low specificity in tuberculosis endemic areas [3]. FDG-PET has given false-negative results for well-differentiated pulmonary adenocarcinoma [4,5], or small volumes of metabolically active tumors [6], and false-positive results for inflammatory nodules [7,8]. FDG-PET/CT is not used in many hospitals due to its high cost, radiation exposure, and the use of a contrast medium.

Magnetic resonance imaging (MRI) is more useful than CT for the visualization of the heart, mediastinal vessels and the pericardium [9]. MRI has an advantage specifically for investigating an invasion into the left atrium via pulmonary veins, the superior vena cava, or myocardium. Diffusion-weighted magnetic resonance imaging (DWI) is applied universally to demonstrate the decreased diffusion of water molecules. DWI is characterized as the random motion of water molecules in biologic tissue, which is called the Brownian movement [10]. DWI was performed first in brain imaging, mainly for the assessment of acute ischemic stroke, intracranial tumors, and demyelinating diseases [11,12]. The diffusion of water molecules in malignant neoplasms is usually decreased compared to that in benign tissue, resulting in a lower apparent diffusion coefficient (ADC) value [13,14]. The MR signal intensity of pulmonary cancers increased drastically compared to that of benign lesions [15]. DWI can distinguish benign from malignant lesions in many organs, especially in the lung [16,17,18,19], in the prostate [20], in the breast [21], and in the liver [22]. Two meta-analyses of MRI (DWI) reported that MRI (DWI) is valuable for a differential diagnosis of benign and malignant pulmonary nodules and masses (PNMs) [23,24].

Although DWI can be valuable for differential diagnosis of lung cancer from benign PNMs, the diagnostic capability may not be perfect. DWI has a weak point for pulmonary abscesses and mycobacteria infections [25]. The usefulness of T2WI was demonstrated, especially in the evaluation of high-intensity fluid materials associated with the organ lesions, such as intratumoral necrosis, cysts, mucus, hemorrhage, or edema [26,27]. Combined assessment of DWI and T2WI works well together for detecting PNMs. We reported MRI (DWI + T2WI) was useful for the assessment of PNMs in a previous paper [25]. In this paper, we compared diagnostic performance between MRI (DWI + T2WI) and FDG-PET/CT.

The purpose of this study was to compare the diagnostic efficacy of FDG-PET/CT and MRI with DWI and T2WI in discriminating malignant from benign PNMs.

## 2. Materials and Methods

### 2.1. Eligibility

The institutional ethical committee of Kanazawa Medical University consented to the study protocol for evaluating FDG-PET/CT and MRI in patients with PNMs (the consented number: No. I302). An informed consent document for the MRI was obtained from each patient after discussing the risks and benefits of the examinations. The study was performed according to the guidelines of the Declaration of Helsinki.

### 2.2. Patients

Patients who had lung cancer or a benign pulmonary nodule and mass (BPNM) in chest X-rays were examined first by chest CT with contrast media. PNMs that were less than 6 mm of solid nodules or 15 mm of part-solid nodules were followed by CT, FDG-PET/CT or MRI for two years. When growth was detected, surgical resection of them was performed. In the patients who had primary lung cancers or BPNMs in CT and had FDG-PET/CT and MRI examinations from May 2009 to April 2020, 331 patients qualified for detailed analysis of FDG-PET/CT and MRI with DWI and T2WI before pathological diagnosis and bacterial diagnosis. Patients in the study had PNMs with a maximum size of 150 mm or less (range 5–150 mm, mean 31.9 mm) in CT, which had no definitive calcification. Patients with a part-solid PNM were included. Lung cancers with pure ground-glass-nodules (GGNs) were excluded. Patients who received prior treatment were excluded. Most of the PNMs were pathologically determined by surgical resection or bronchoscopic examination. The other PNMs were determined by bacterial culture or a roentgenographically follow-up study. The PNMs were determined as benign when the PNMs decreased in size or disappeared upon review of chest X-rays films or CT. Out of 331 patients, 3 patients were excluded because of insufficient data. Finally, 328 PNMs were registered in the study (Table 1), of which 208 patients were men and 120 were women. Their mean age was 68.3 years old (range 37 to 85). There were 278 lung cancers and 50 BPNMs. Twenty-nine patients had part-solid PNMs. Out of the 328 patients with PNMs, 311 were also used in another paper [25]. The diagnosis was made pathological in all 278 lung cancers. The 278 lung cancers consisted of 192 adenocarcinomas, 64 squamous cell carcinomas, 5 large cell neuroendocrine carcinomas (LCNECs), 3 large cell carcinomas, 4 adenosquamous carcinomas, 2 carcinoids, 7 small cell carcinomas and 1 carcinosarcoma. TNM classification and the lymph node stations of lung cancer were classified according to the new definitions in UICC 8 [28]. There were 2 pathological T1mi (pT1 mi) carcinomas, 69 pT1a carcinomas, 53 pT1b carcinomas, 5 pT1c carcinomas, 80 pT2a carcinomas, 22 pT2b carcinomas, 39 pT3 carcinomas, and 8 pT4 carcinomas. There were 222 pathological N0 (pN0) carcinomas, 34 pN1 carcinomas, and 22 pN2 carcinomas. There were 269 pathological M0 (pM0) carcinomas, 6 pM1a carcinomas, 2 pM1b carcinomas, and 1 M1c carcinoma. There were 122 pStage IA carcinomas, 60 pStage IB carcinomas, 28 pStage IIA carcinomas, 26 pStage IIB carcinomas, 32 pStage IIIA carcinomas, 1 pStage IIIB carcinoma, 8 pStage IVA carcinomas, and 1 pStage IVB carcinoma.

For 50 BPNMs, there were 39 inflammatory BPNMs [Mycobacterial disease 13 (tuberculosis 5, nontuberculous mycobacteria 8), pneumonia 12, pulmonary abscess 7, pulmonary scar 3, organized pneumonia 2, pulmonary granuloma 1 and sarcoidosis 1, and 11 non-inflammatory BPNMs (hamartoma 5, pulmonary sequestration 2, nodular lymphoid hyperplasia 1, inflammatory myofibroblastic tumor 1, encapsulated pleural effusion 1], and pleural cyst 1). Twenty-nine BPNMs were pathologically diagnosed by resection. Three BPNMs were diagnosed as a mycobacterial disease by bacterial culture. The remaining 18 BPNMs were diagnosed as benign diseases by reduced size or vanishing of the BPNMs.

### 2.3. FDG-PET/CT

FDG-PET/CTs were performed using a PET camera (SIEMENS Biograph Sensation 16, Erlangen, Germany). Each patient fasted at least 6 h before being scanned. The plasma glucose level was checked before the injection of FDG and was confirmed to be <180 mg/dL in all patients. The dose of ^18^F-FDG was 3.7 MBq/Kg of body weight. The CT data were collected in the helical acquisition mode. Each patient’s CT images were matched to the pixel size of his or her PET data in order to match the in-slice resolution of the PET images. A 60-min uptake period is needed to allow the contrast medium to take effect, and an emission scan was taken for 3 min per bed position, and whole-body scanning was performed from head to pelvis. The PET data were reconstructed with an ordered-subset expectation algorithm using the CT images for attenuation correction. The PET-CT images were analyzed on a dedicated workstation.

After image reconstruction, a round region of interest (ROI) of a 2-dimension (2D) was drawn on the fused CT image by the radiologist with 30 years of radioisotope scintigraphy and FDG-PET/CT experience who was not aware of the patients’ clinical data. For the lesions with negative or faintly positive FDG-PET findings, the region of interest (ROI) was drawn on the fusion image with the corresponding CT. From those ROIs, the maximum standardized uptake value (SUVmax) was obtained.

### 2.4. MR Imaging

All MR images were performed without contrast media using a 1.5 T magnetic scanner (Magnetom Avanto, Siemens, Erlangen, Germany). The conventional MR images were made of axial and coronal T1 weighted turbo spin-echo (TSE) and T1 gradient recalled echo (GRE), and axial and coronal T2-weighted TSE (Table 2). Examination of the 1.5-T MRI was conducted as follows: T2-weighted imaging (T2WI) was performed in a TSE; TR/TE, 4400–6000/74 ms; FOV, 350 × 240 mm; matrix, 320 × 198; thickness, 6.0 mm), Flip angle 90°. T1-weighted imaging (T1WI) was performed in a TSE and a GRE. DWIs with a single-shot echo-planar method were conducted with a slice thickness of 6 mm under SPAIR (spectral attenuated inversion recovery) with a respiratory triggered scan with the next condition: TR/TE/flip angle, 3000–4500/65/90; diffusion gradient encoding in three orthogonal directions; b-value = 0 and 800 s/mm^2^; field of view, 350 mm; matrix size, 128 × 128.

For the visual detection in DWI, diffusion detectability scores (DDSs) of lung cancers and BPNMs were determined visually on a 5-point scale in our article [29], which was a revision of the Hahn SY model [30]. After image reconstruction, a two-dimensional (2D) round or elliptical region of interest (ROI) was drawn on the lesion that was detected visually on the ADC map with reference to T2-weighted or CT image. The procedures were repeated three times, and the minimum ADC value was obtained. The T2 contrast ratio (T2 CR) of a PNM was defined based on the definition of Koyama et al. [31]: T2 CR = the ratio of T2 signal intensity of a PNM divided by T2 signal intensity of the rhomboid muscle. T2 signal intensities of PNMs were obtained by drawing round, elliptical, or free-hand ROIs on lesions that were detected visually on the T2WI. The ROI drawn on the muscle was fixed at 120 mm^2^ (a round of 8 mm in size) according to the description of Koyama et al. The MRI data were evaluated by a radiologist (M.D.) with 25 years of MRI experience who was unaware of the patients’ clinical data and a pulmonologist (K.U.) with 28 years of experience. The experienced author (K.U.) performed all measurements, supported by the experienced radiologist (M.D.). They eventually reached the same consensus. There was no discrepancy in the data between the radiologist and the pulmonologist.

### 2.5. PET and MRI Analysis

In FDG-PET/CT, the receiver operating characteristics (ROC) curve of the diagnostic performance of SUVmax for discriminating BPNM from lung cancer was obtained, and sensitivity, specificity, and accuracy by the optimal cutoff values (OCV) were determined. The mean SUVmax of lung cancer was compared to that of BPNM.

In MRI, relationships between DDSs and lung cancer/BPNM were shown. The ROC curve of the diagnostic performance of ADC for discriminating BPNM from lung cancer was obtained, and sensitivity, specificity, and accuracy by the OCV were determined. The mean ADC of lung cancer was compared to that of BPNM. The ROC curve of the diagnostic performance of T2 CR for discriminating BPNM from lung cancer was obtained, and sensitivity, specificity and accuracy by the OCV were determined. The mean T2 CR of lung cancer was compared to that of BPNM.

Diagnostic performance of SUVmax, ADC, and T2 CR were compared between lung cancer and BPNM.

### 2.6. Statistical Analysis

The data are presented as the mean ± standard deviation. A non-parametric test (Mann–Whitney U test) was applied to compare the mean value of the two groups. A Chi-square test was used for the comparison of ratios. A ROC curve was applied to evaluate the diagnostic capability of SUVmax, ADC value and T2 CR value in terms of malignant–benign differentiation. The OCV of SUVmax, ADC, and T2 CR for a differential diagnosis were determined using GraphPad Prism (Version 5.02, GraphPad Software, Inc., La Jolla, CA, USA). PNMs with an SUVmax of the OCV or more were defined as positive. PNMs with an SUVmax less than the OCV or those which could not be detected on FDG-PET were defined as negative. PNMs with an ADC of the OCV or less were defined as positive. PNMs with an ADC more than the OCV were defined as negative. PNMs with a T2 CR of the OCV or less were defined as positive. PNMs with a T2 CR or more than the OCV were defined as negative. The sensitivity, specificity, and accuracy of SUVmax versus ADC or T2 CR for PNMs were compared using the McNemar test. The statistical analyses were performed based on StatView for Windows (Version 5.0; SAS Institute Inc., Cary, NC, USA). A *p*-value of <0.05 was defined as statistically significant.

## 3. Results

### 3.1. Radiological Characteristics Based on DDSs of DWI

Relationships between DDSs and lung cancer/BPNM were presented in Table 3. In lung cancer cases, 209 PNMs (75.2%) were classified in DDS5 and 32 PNMs (11.5%) in DDS4. As a result, 241 PNMs (86.7%) were classified in DDS4 and more. In BPNMs, pulmonary abscesses and mycobacterial infections showed decreased diffusion. 22 BPNMs (44.0%) were classified in DDS5 and 14 (28.0%) in DDS2. The mean DDS of lung cancers (4.55 ± 0.92) was significantly higher than that (3.77 ± 1.32) of BPNMs (*p* < 0.0001) (Figure 1).

### 3.2. Radiologic Presentations of CT, FDG-PET/CT, DDS of DWI, ADC Map and T2WI in PNMs

According to the DDSs, malignant/benign PNMs, chest CT, FDG-PET/CT, DWI, ADC map, and T2WI are presented in Figure 2, Figure 3, Figure 4, Figure 5 and Figure 6. 

### 3.3. ROC Analysis and Diagnostic Performance of SUVmax, ADC and T2 CR

The ROC curve of the diagnostic performance of SUVmax for discriminating BPNM from lung cancer presented area under the ROC curve (AUC) was 67.5% (Figure 7). When the OCV of SUV max was set at 3.605, the sensitivity was 65.25%, the specificity 60.0%, and the accuracy 64.3%.

Detail of SUVmax, ADC, and T2 CR between lung cancer and BPNM were shown in Table 4. In relationships between the mean SUVmax and lung cancer/BPNM, the mean SUVmax (7.89 ± 6.73) of lung cancer was significantly higher than that (4.11 ± 3.90) of BPNM (*p* < 0.0001) (Figure 8, Table 4).

The ROC curve of the diagnostic performance of ADC for discriminating lung cancer from BPNM showed the AUC was 74.3% (Figure 9). When the OCV of ADC was set at 1.459 × 10^−3^ mm^2^/s, the sensitivity was 84.0%, the specificity 64.0%, and the accuracy 81.1%. In relationships between the mean ADC and lung cancer/BPNM, the mean ADC (1.24 ± 0.29 × 10^−3^ mm^2^/s) of lung cancer was significantly lower than that (1.67 ± 0.59 × 10^−3^ mm^2^/s) of BPNM (*p* < 0.0001) (Figure 10, Table 4).

The ROC curve of the diagnostic efficacy of T2 CR for differentiating lung cancer from BPNM the AUC was 72.4% (Figure 11). When the OCV of T2 CR was set at 2.46, the sensitivity was 87.3%, the specificity 64.0%, and the accuracy 83.8%. In relationships between the mean T2 CR and lung cancer/BPNM, the mean T2 CR (2.05 ± 0.53) of lung cancer was significantly lower than that (2.73 ± 1.04) of BPNM (*p* < 0.0001) (Figure 12, Table 4).

### 3.4. Comparison of Diagnostic Performance of SUVmax, ADC and T2 CR

When the OCVs were set at 3.605 for SUVmax, 1.459 × 10^−3^ mm^2^/s for ADC, and 2.46 for T2 CR, sensitivity, specificity and accuracy were calculated using the McNemar test (Table 5). Concerning comparison of sensitivity among SUVmax ADC and T2 CR, the sensitivity (0.658 (183/278)) of SUVmax was significantly lower than that (0.838 (233/278)) of ADC (*p* < 0.001), and significantly lower than that (0.871 (242/278)) of T2 CR (*p* < 0.001). Concerning the comparison of specificity among SUVmax, ADC, and T2 CR, the specificity (0.620 (31/50)) of SUVmax was the same as that (0.640 (32/50)) of ADC (not significant) and the same as that (0.640 (32/50)) of T2 CR (not significant). Concerning the comparison of accuracy among SUVmax, ADC, and T2 CR, the accuracy (0.652 (214/328)) of SUVmax was significantly lower than that (0.808 (265/328)) of ADC (*p* < 0.001) and significantly lower than that (0.835 (274/328)) of T2 CR (*p* < 0.001).

## 4. Discussion

Our results showed that the sensitivity (0.658) of SUVmax was significantly lower than that (0.838) of ADC (*p* < 0.001) and that (0.871) of T2 CR (*p* < 0.001). The specificity (0.620) of SUVmax was the same as that (0.640) of ADC and as that (0.640) of T2 CR. The accuracy (0.652) of SUVmax was significantly lower than that (0.808) of ADC (*p* < 0.001) and that (0.835) of T2 CR (*p* < 0.001). This result showed MRI with DWI and T2 WI is an alternative to FDG-PET/CT. In this study, T2 CR of T2WI had a good quality for differential diagnosis of PNMS. T2WI can make up for a weak point with DWI. A pulmonary abscess that strongly presents decreased diffusion in DWI could be differentiated from lung cancers using T2WI [32]. Combined analysis of DWI and T2WI could judge PNMs more precisely and would be more accurate for differential diagnosis of PNMs [25]. Ultimately this study shows MRI can replace PET-CT for differential diagnosis of PNMs. Adding DWI to T2WI was reported to be helpful for detecting viable tumors after neoadjuvant chemoradiotherapy compared with T2WI alone or FDG-PET/CT in patients with locally advanced rectal cancer [33].

Concerning the comparison between DWI and FDG-PET/CT, DWI was described to be more useful than FDG-PET/CT in the diagnosis of primary pulmonary lesions and the nodal assessment of non-small cell lung cancers (NSCLCs) [18,34]. The advantage of DWI can be explained not only by DWI having fewer false-positive results for N staging of NSCLC compared with FDG-PET [35] but also by DWI having fewer false-negative results [18]. Two articles described the diagnostic ability of DWI with that of FDG-PET/CT for PNMs [16,36]. The sensitivity and the accuracy of FDG-PET/CT were significantly lower [16], or the sensitivity of FDG-PET/CT was significantly lower [36] than those of DWI, which was identical to the result of this study.

Koyama et al. [31] reported that MRI can detect and stage lung cancer, and this method could be an excellent alternative to CT or PET/CT in the investigation of pulmonary malignancies and other diseases [37]. Conventional MRI can reveal the essential differences between mass-like tuberculosis and lung cancer and may be helpful for discriminating pulmonary masses [38]. When an invasion is unclear by CT criteria, MRI can play an important role in defining lesser degrees of invasion [39]. MRI is superior to CT for the visualization of the pericardium, the heart and mediastinal vessels [40]. MRI can be of use specifically for assessing invasion of the myocardium, superior vena cava, or extension of the tumor into the left atrium via pulmonary veins [40]. Although FDG-PET/CT is thought to be more effective for this purpose, MRI has the advantage of being more universally available and less expensive [37].

Pure bronchioloalveolar carcinoma (BAC) is a subtype of adenocarcinoma, which appears as lepidic growth of tumor cells along the alveoli without vascular, stromal, lymphatic, or pleural invasion [41], and appears as pure ground-glass-nodule (GGN) on CT scans. The SUVmax of GGN-type lung cancers was described to be 0.64 ± 0.19 [42]. Adenocarcinomas with BAC features have been rapidly increasing in incidence in the past two decades [43]. Although Could MK et al. [2] described meta-analysis results that presented sensitivity by FDG-PET was over 90% for malignant pulmonary lesions, these results were from studies released from January 1966 to September 2000 in the MEDLINE and CANCERLIT databases, and they were mainly solid lung cancers, whose FDG uptake was higher than pure BACs and adenocarcinomas of predominantly BAC features. Nowadays, CTs are performed widely and cases with pure BACs, adenocarcinomas of predominantly BAC features, or tiny lung cancers within 10 mm have increased. They seem to be false-negatives in FDG-PET/CT owing to their low-level metabolism and tiny metabolically active tumors. For diagnosis of non-solid solitary pulmonary nodules, the cutoff of 1.5 was applied for SUVmax [44]. Recently, the sensitivity by FDG-PET for malignant pulmonary lesions has lowered due to the fact that adenocarcinomas with BAC features have been rising in incidences in the past two decades [43]. One of the reasons for the lower sensitivity (0.658) of PNMs on FDG-PET in this study was guessed to be associated with increased adenocarcinomas with predominantly BAC features.

For contrast-enhanced CT, PNMs that can be enhanced by more than 20 Hounsfield units (HU) after the administration of contrast medium was usually malignant, whereas PNMs that can be enhanced less than 15 HU were benign [45]. A recent meta-analysis of ten contrast-enhanced CT studies presented a pooled sensitivity of 93%, a specificity of 76%, a positive predictive value (PPV) of 80%, and a negative predictive value (NPV) of 95% for PNMs [46], and the data sources were studies published in PubMed between January 1990 and December 2005. Most PNMs of this study were solid solitary pulmonary nodules. Concerning the comparison between CT and FDG-PET/CT, the sensitivity and specificity for CT were 0.94 (95% confidence interval (CI): 0.87–0.97), 0.73 (95% CI: 0.64–0.80), and the pooled sensitivity and specificity for FDG-PET/CT were 0.89 (95% CI: 0.85–0.92), 0.78 (95% CI: 0.66–0.86) [47]. No significant differences were observed between CT and FDG-PET/CT for sensitivity, specificity [47]. The data sources were studies published between January 1992 and 2018. Most PNMs of this study were solid solitary pulmonary nodules. These results were better than those of this study that included part-solid PNMs.

Mark L. Schiebler, in the Editorial of Radiology in 2016, cited our paper on whole-body DWI MRI (DWIBS) for lung cancer as follows [48]. There is a single paper by Usuda et al. [49] that presents that whole-body DWI MRI can be performed to adequately stage NSCLC. He described that if the diagnostic ability of whole-body DWI MRI is proved to be equivalent to PET-CT for clinical staging of lung cancer while also reducing medical costs, whole-body DWI MRI will ultimately replace FDG-PET/CT in the future. In other organs, whole-body DWI MRI is a valid technique for the assessment of bone marrow involvement in lymphoma patients and is more efficient than FDG PET/CT for the assessment [50]. Whole-body DWI MRI is a sensitive and specific imaging technique for lymphoma evaluation, supporting its use in place of CE-CT for staging [51].

The use of radiomics in the differential diagnosis between benign and malignant PNMs will be a great tool for the future. A large number of indeterminate pulmonary nodules and masses provides considerable diagnostic and management challenges. Conventional nodule evaluation relies on visually identifiable discriminators such as size and speculation. Radiomics is a developing field aimed at deriving automated quantitative imaging features from medical images that can predict nodule and tumor behavior non-invasively. In CT or FDG-PET/CT, radiomics has been extensively applied to lung cancer and multiple studies evaluated its role in diagnosis, prognosis, and response to treatment [52]. In MRI, there is also the possibility that radiomics is useful for diagnosis, prognosis, and response to treatment of lung cancer. Concerning the use of radiomics in the differential diagnosis between benign and malignant lung nodules, ADC histograms of PNMs are efficient methods for differential diagnosis [53].

When a PNM could not be judged as malignant or benign in CT, we should examine it with MRI for the assessment. When we obtain a strong diffusion in which ADC is lower than its own OCV of the PNMs, the PNM must be lung cancer or a pulmonary abscess or a mycobacterial infection with abscess. Additional T2WI can prove it is lung cancer when its T2 CR is lower than its own OCV of the PNMs and can prove it is a pulmonary abscess or a mycobacterial infection when its T2 CR is higher than its own OCV of the PNMs.

Limitations of FDG-PET/CT were radiation exposure, the need for contrast medium, a 6-h fast before FDG-PET/CT, the limitation for patients with diabetes mellitus and an expensive cost. The limitations of MRI are the impossibility for patients with metal medical devices, pacemakers, or tattoos. The advantages of DWI are easier accessibility, relatively cheaper, and no X-rays radiation exposure compared with PET-CT. The number of hospitals where PET-CT is equipped is limited due to the difficulty in handling the radioisotope of ^18^F-FDG. The cost of DWI is almost one-third of that of a PET-CT examination. In addition, no radiation exposure during an MRI examination is favorable compared to some radiation exposure during a PET-CT examination.

There are two disadvantages of DWI. First, benign PNMs accompanied by histopathological necrosis such as a pulmonary abscess or mycobacterial infection show restricted diffusion and lower ADC values. Abscesses and thrombi impede the diffusion of water molecules owing to their hyperviscous characteristics [54,55]. The pus itself causes low ADC values and heavily impedes water mobility, and the pus may be associated with its high cellularity and viscosity [56]. In the assessment of DWI, 22% of benign lesions express restricted diffusion with high b-values [57]. These papers can explain some BPNMs were false-positive when DWI was applied for the assessment of BPNMs with abscesses.

Second, mucinous adenocarcinomas are hypointense in DWI and had higher ADC values, which could be misjudged as benign lesions in DWI. Mucinous carcinomas possess higher ADC values and a lower DWI signal intensity than tubular adenocarcinoma in the ano-rectal region because mucinous carcinomas possess rather lower cellularity than tubular adenocarcinomas [58]. Mucinous adenocarcinomas will be also misdiagnosed as benign lesions in T2WI because they contain a large quantity of viscous liquid [25].

We have to keep in mind that the research had two limitations. First, it was a retrospective research project and was conducted at a single institution. The number of benign PNMs was only 50. For a more accurate assessment, additional cases of BPNM are necessary. Further, adequately powered prospective randomized trials will be needed to evaluate FDG-PET/CT and MRI for discriminating between lung cancer and BPNM.

## 5. Conclusions

The purpose of this study was to compare the diagnostic efficacy of FDG-PET/CT and MRI with T2WI and DWI in distinguishing malignant from benign PNMs. There were 278 lung cancers and 50 BPNMs. The sensitivity and accuracy of DWI and T2WI in MRI were significantly higher than those of FDG-PET/CT. Ultimately MRI can replace FDG-PET/CT for differential diagnosis of PNMs saving healthcare systems money while not sacrificing the quality of care.

## Figures and Tables

**Figure 1 cancers-13-05166-f001:**
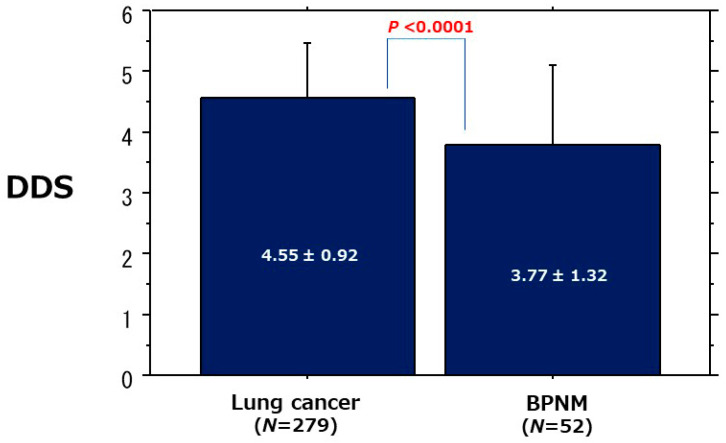
Comparison of DDS between lung cancer and BPNM. The mean DDS (4.55 ± 0.92) of lung cancer was significantly higher than that (3.77 ± 1.32) of BPNM (*p* < 0.0001).

**Figure 2 cancers-13-05166-f002:**
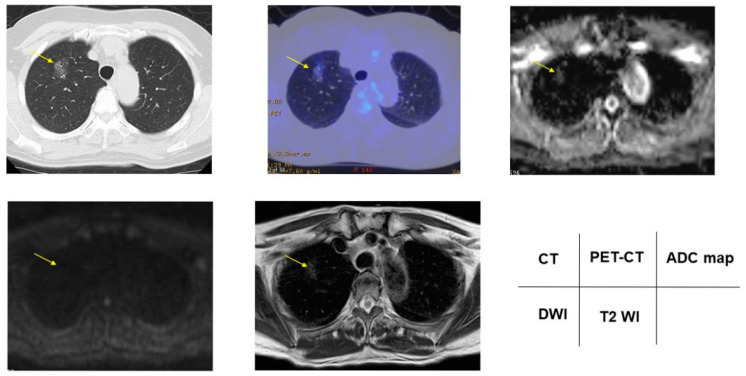
Papillary adenocarcinoma showed 1.90 for SUVmax, DDS 1, 1.85 × 10^−3^ mm^2^/s for ADC, and 1.97 for T2 CR. Yellow arrows shows PNMs.

**Figure 3 cancers-13-05166-f003:**
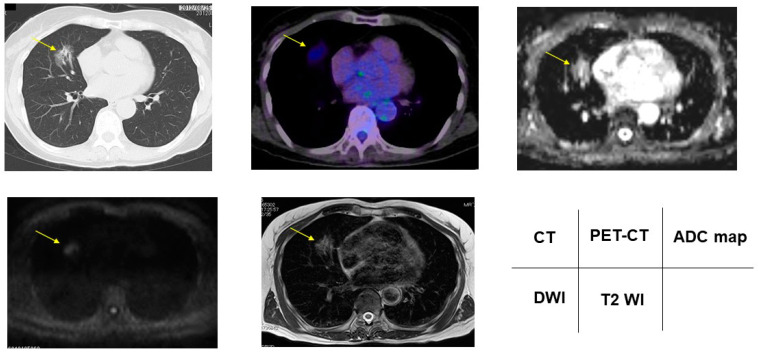
Acinar adenocarcinoma showed 2.17 for SUVmax, DDS 2, 1.67 × 10^−3^ mm^2^/s for ADC, and 1.34 for T2 CR. Yellow arrows shows PNMs.

**Figure 4 cancers-13-05166-f004:**
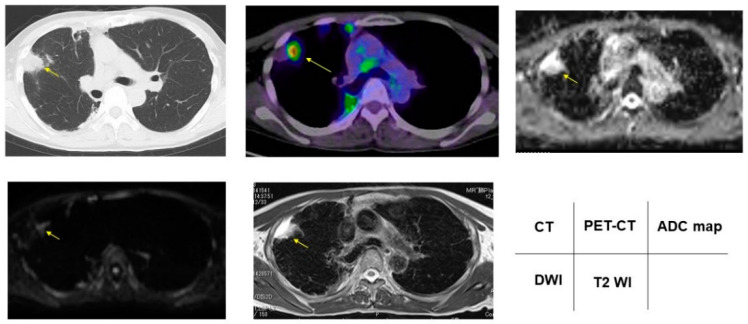
Nontuberculous mycobacteria showed 5.00 for SUVmax, DDS 3, 2.50 × 10^−3^ mm^2^/s for ADC, and 4.00 for T2 CR. Yellow arrows shows PNMs.

**Figure 5 cancers-13-05166-f005:**
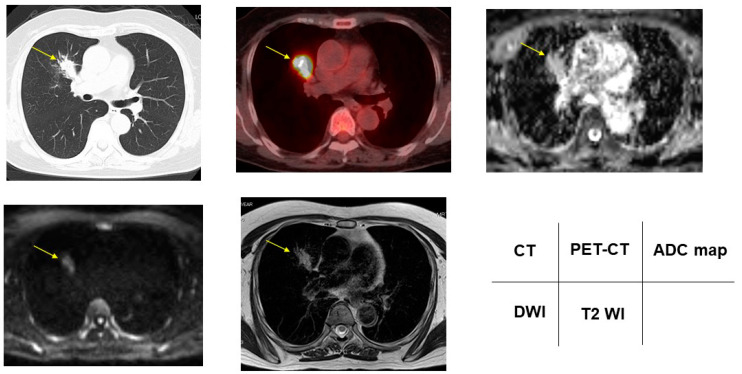
Pneumonia showed 15.0 for SUVmax, DDS 4, 1.62 × 10^−3^ mm^2^/s for ADC, and 3.33 for T2 CR. Yellow arrows shows PNMs.

**Figure 6 cancers-13-05166-f006:**
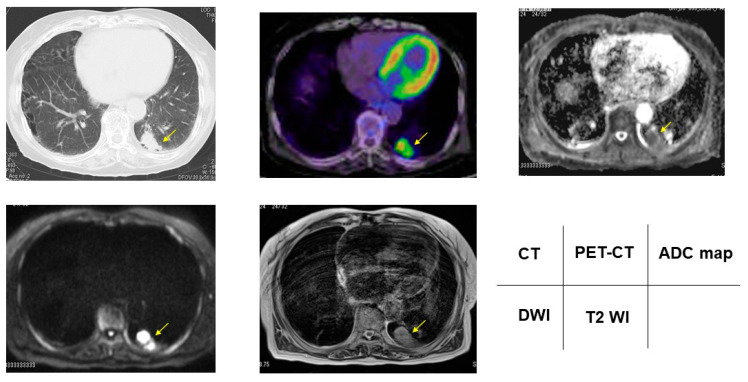
Papillary adenocarcinoma showed 7.7 for SUVmax, DDS 5, 1.20 × 10^−3^ mm^2^/s for ADC, and 1.98 for T2 CR. Yellow arrows shows PNMs.

**Figure 7 cancers-13-05166-f007:**
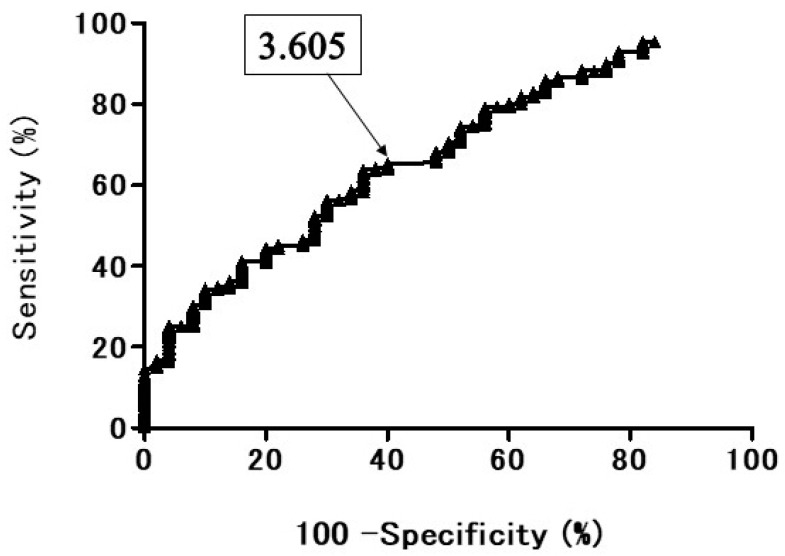
The receiver operating characteristic (ROC) curve presents the diagnostic performance of SUVmax for distinguishing the benign pulmonary nodule and mass (BPNM) from lung cancer. The area under the ROC curve 67.5%. SUV max = 3.605, sensitivity 65.25%, specificity 60.0%, accuracy 64.3%.

**Figure 8 cancers-13-05166-f008:**
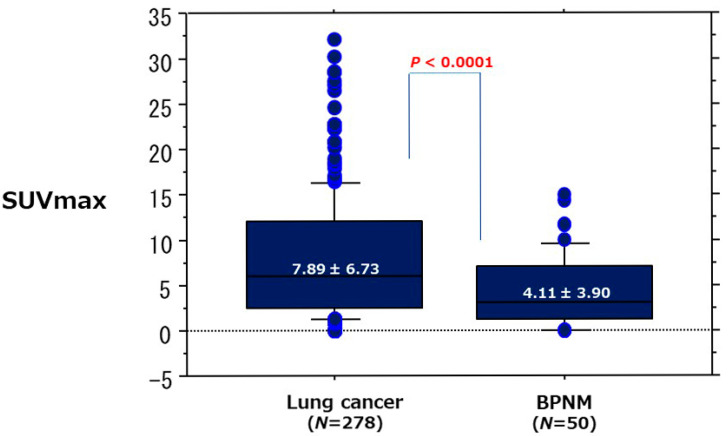
Comparison of SUVmax between lung cancer and BPNM. The mean SUVmax (7.89 ± 6.73) of lung cancer was significantly higher than that (4.11 ± 3.90) of BPNM (*p* < 0.0001).

**Figure 9 cancers-13-05166-f009:**
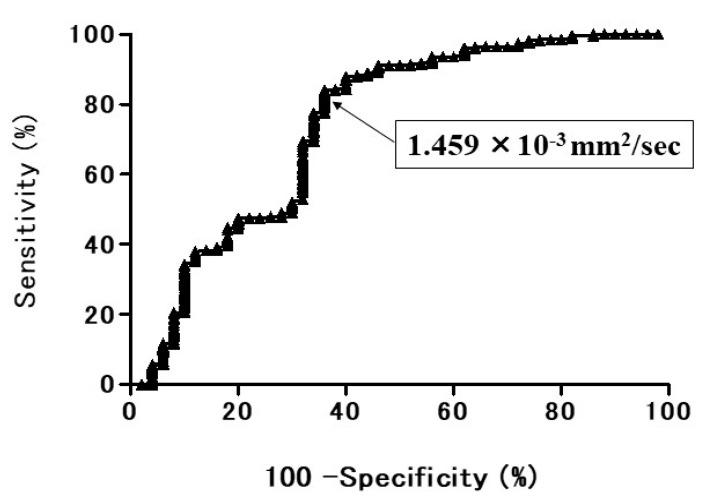
The receiver operating characteristic (ROC) curve shows the diagnostic performance of ADC for distinguishing the benign pulmonary nodule and mass (BPNM) from lung cancer. The area under the ROC curve 74.3%. ADC = 1.459 × 10^−3^ mm^2^/s, sensitivity 84.0%, specificity 64.0%, and accuracy 81.1%.

**Figure 10 cancers-13-05166-f010:**
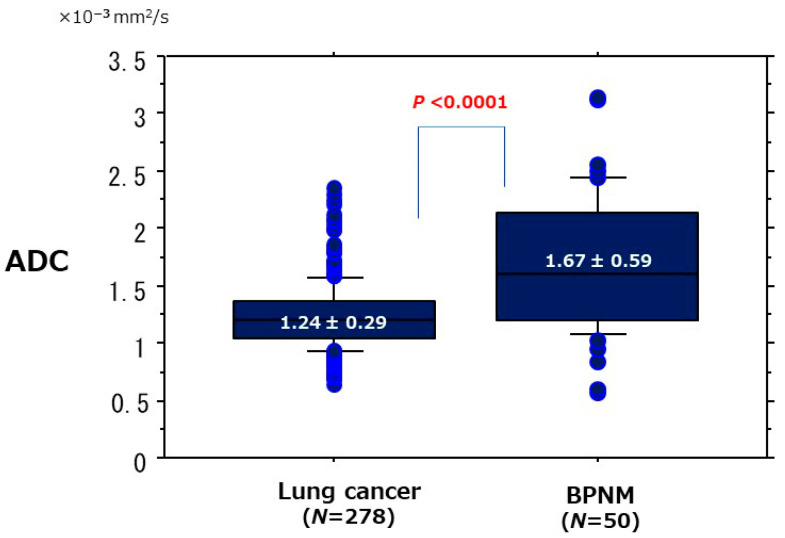
Relationships between the mean ADC and lung cancer/BPNM. The mean ADC (1.24 ± 0.29 × 10^−3^ mm^2^/s) of lung cancer was significantly lower than that (1.67 ± 0.59 × 10^−3^ mm^2^/s) of BPNM (*p* < 0.0001).

**Figure 11 cancers-13-05166-f011:**
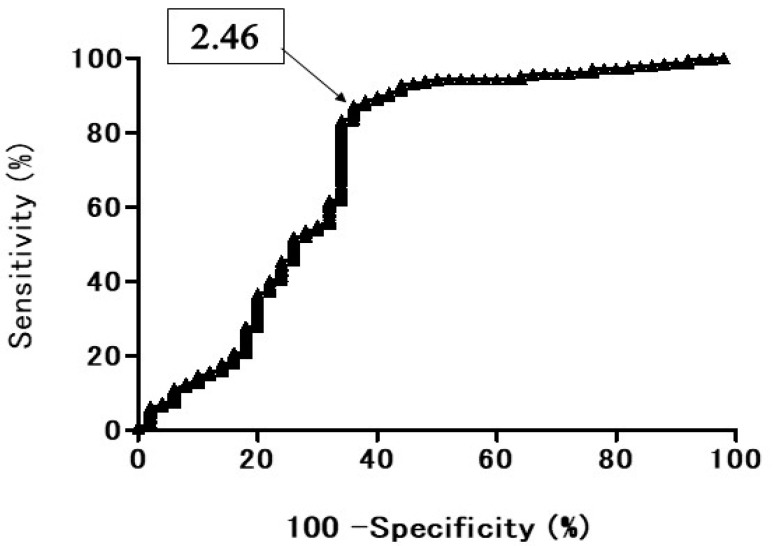
The receiver operating characteristic (ROC) curve shows the diagnostic performance of T2 CR for distinguishing the benign pulmonary nodule and mass (BPNM) from lung cancer. The area under the ROC curve 72.4%. T2 CR = 2.46, sensitivity 87.3%, specificity 64.0%, and the accuracy 83.8%.

**Figure 12 cancers-13-05166-f012:**
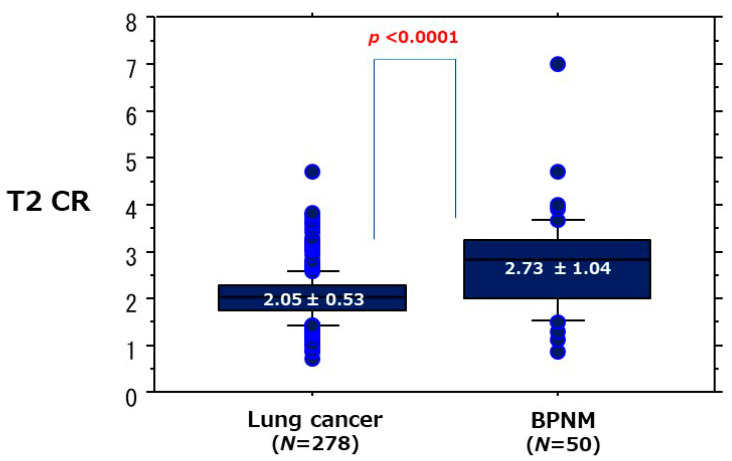
Relationships between the mean T2 CR and lung cancer/BPNM. The mean T2 CR (2.05 ± 0.53) of lung cancer was significantly lower than that (2.73 ± 1.04) of BPNM (*p* < 0.0001).

**Table 1 cancers-13-05166-t001:** Patients’ characteristics.

Diagnosis		No. of Patients	
Lung cancer		278	
	adenocarcinoma		192
	squamous cell ca.		64
	LCNEC		5
	Large cell ca.		3
	Adenosquamous ca.		4
	Carcinoid		2
	Small cell ca.		7
	Carcinosarcoma		1
Inflammatory benignity		39	
	Mycobacterial disease		13 (Tbc 5, NTM 8)
	Pneumonia		12
	Pulmonary abscess		7
	Pulmonary scar		3
	Organized pneumonia		2
	Other		2
Non-inflammatory benignity		11	
	Hamartoma		5
	Pulmonary sequestration		2
	Other		4
Total No. of patients		328	

LCNEC: large cell neuroendocrine carcinoma, Tbc: tuberculosis, NTM: nontuberculous mycobacteria.

**Table 2 cancers-13-05166-t002:** Imaging parameters used for the study on a 1.5 T magnetic resonance scanner.

Sequence	Echo Time (TE) (ms)	Repetition Time (TR) (ms)	Slice Thickness (mm)	Field of View (FOV) (mm)	Matrix Size
T1 turbo-spin echo (TSE)	5.4	600–1000	6 mm	320 × 198	320 × 198
T1 gradient recalled echo (GRE)	4.78	6.54	3.5 mm	380 × 240	256 × 151
T2 turbo-spin echo (TSE)	74	4400–6000	6 mm	350 × 240	320 × 198
DWI SPAIR with respiratory triggered fat suppression	65	3000–4500	6 mm	350	128 × 128

SPAIR: spectral attenuated inversion recovery.

**Table 3 cancers-13-05166-t003:** Relationships between diffusion detectability scores (DDSs) and lung cancer /BPNMs.

Degree of DDS	DDS1	DDS2	DDS3	DDS4	DDS5	No. of Total Cases
Lung cancer	4 (1.4%)	14 (5.0%)	19 (6.8%)	32 (11.5%)	209 (75.2%)	278 (100%)
BPNM	0	14 (28.0%)	7 (14.0%)	7 (13.5%)	22 (44.0%)	50 (100%)
No. of total cases	4	28	26	39	231	328

**Table 4 cancers-13-05166-t004:** SUVmax, ADC, and T2 CR between lung cancer and BPNM.

Variable	Lung Cancer				BPNM				*p*-Value
	Mean	Standard Deviation	Minimum	Maximum	Mean	Standard Deviation	Minimum	Maximum	
SUVmax	7.89	6.73	0	32	4.11	3.9	0	15	*p* < 0.0001
ADC	1.24	0.29	0.646	2.355	1.67	0.59	0.556	3.13	*p* < 0.0001
T2 CR	2.05	0.53	0.703	4.707	2.73	1.04	0.877	7.019	*p* < 0.0001

**Table 5 cancers-13-05166-t005:** Sensitivity, specificity, and accuracy of PNMs using their OCVs.

Variable	TP	FN	TN	FP	Sensitivity	Specificity	Accuracy
SUVmax	183	95	31	19	0.658 (183/278) *	0.62 (31/50) *	0.652 (214/328) *
ADC	233	45	32	18	0.838 (233/278) **	0.64 (32/50) **	0.808 (265/328) **
T2 CR	242	36	32	18	0.871 (242/278) ***	0.64 (32/50) ***	0.835 (274/328) ***
					* vs. ** *p* = 0.0089	* vs. ** N.S.	* vs. ** *p* = 0.0365
					* vs. *** *p* = 0.0176	* vs. *** N.S.	* vs. *** *p* < 0.001

TP: true positive, FN: false negative, TN: true negative, FP: false positive, N.S.: not significant. Asterisks (*, ** or ***) shows the special values of the columns.

## Data Availability

The data presented in this study are available in this article.

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
