# Peer review of "Pulmonary Nodule and Mass: Superiority of MRI of Diffusion-Weighted Imaging and T2-Weighted Imaging to FDG-PET/CT"

_cancers, 2021, doi:10.3390/cancers13205166_

Round 1

Reviewer 1 Report

The authors addressed all my questions.

Author Response

Thank you for reviewing our paper.

Reviewer 2 Report

General Comments:

The manuscript has been improved according to reviewers’ suggestions. It lacks some details before publication.

Please see specific comments.

Specific comments:

Materials and Methods:

  • How many patients underwent surgery and had histologic diagnosis? Please specify.

Discussion:

  • Overall ok, however you might consider to avoid the division into paragraph in this section.

Tables and Figures:

  • Overall please remove the annotations from medical images.

Author Response

Answers to Comments and Suggestions for Authors

General Comments:

The manuscript has been improved according to reviewers’ suggestions. It lacks some details before publication.

Please see specific comments.

Specific comments:

Materials and Methods:

How many patients underwent surgery and had histologic diagnosis? Please specify.

The diagnosis was made pathologically in all 278 lung cancers and 29 BPNMs. Three BPNMs were diagnosed as mycobacterial disease by bacterial culture. The remaining 18 BPNMs were diagnosed as benign diseases by reduced size or vanishing of the BPNMs. At the results, 307 PNMs were diagnosed pathologically.

Discussion:

Overall ok, however you might consider to avoid the division into paragraph in this section.

Thank you for your advice.  I avoided the division into paragraph in Discussion section.  I deleted subtitles in Discussion section.

Tables and Figures:

Overall please remove the annotations from medical images.

According to advice of reviewer 1, I pointed out the tumor position in every image.

This manuscript is a resubmission of an earlier submission. The following is a list of the peer review reports and author responses from that submission.

Round 1

Reviewer 1 Report

This paper was to compare the diagnostic efficacy of FDG-PET/CT and MRI with T2WI and DWI in distinguishing malignant from benign PNMs, their conclusions are ultimately MRI can replace FDG-PET/CT for differential diagnosis of PNMs. 

Please address the following questions.

  1. For the title, FDG-PET/CT versus MRI, Should be specified DWI?
  2. For the fig1, in different images, please point out the tumor position in every image.
  3. Fig 8: Sample sizes are very different,  otherwise, error bars are somewhat overlapping. Are you planning to add more BPNM samples to calculations? suggestions: try non-parametric tests or try effect size.

 4.What is the novelty of this paper? If you want to specify about DWI,  you may need to mention it a bit more in the introduction and express it in the title. This paper has some comparisons (https://www.ncbi.nlm.nih.gov/pmc/articles/PMC5003572/) about MRI, but not DWI.

Reviewer 2 Report

General Comments:

The manuscript aims to investigate the ability of MRI in defining the nature of pulmonary nodules in comparison with FDG-PET/CT. 278 patients with lung cancer and 50 with benign lung nodules were retrospectively studied with baseline CT, PET/CT and MRI and images analysed by expert radiologists. MRI resulted to be superior than PET/CT in sensitivity and accuracy. Results demonstrated to be promising but future investigations needed.

The object of the study is very interesting; however, the manuscript has several limitations, in almost all sections, with a specific attention on MM, Results and Discussion sections.

Please see specific comments.

Specific comments:

Title: Please rewrite the Title in a shorter and more catchy manner.

Keywords: Please consider to avoid ADC and pulmonary abscess.

Simple Summary: Please completely rewrite, you should enhance the main results obtained in the study in a clearer manner. In particular, you might avoid numeric results in this summary, considering to introduce the adding value of your paper and some key points. Furthermore, you have to spell out all abbreviations.

Abstract:

  • Overall, the abbreviations must be clarified the first time
  • This section seems to be quite similar to simple summary, please rewrite.
  • Please briefly describe inclusion and exclusion criteria. Statistical analysis performed and the use of contrast media for MRI and CT.
  • Please add the nature of the study.
  • You should describe the main results in a punctual manner, this section sounds overall confusing and lacks a clear aim.
  • Please modify the conclusions, it sounds quite confusing. You should describe some take home messages.

Introduction:

  • Overall, the abbreviations must be clarified the first time.
  • Please provide some details concerning clinical management in case of doubt nature of pulmonary nodules.
  • Please avoid the description of features and strengths of MRI, it seems to be unnecessary in this paper. You might enhance in a stronger manner the literature data about the study based on MRI applied to differential diagnosis of lung nodules. What about diagnostic performance? Accuracy, sensitivity, and specificity?
  • You should describe the adding value of your paper in comparison with previous data.

Materials and Methods:

  • Please be more consistent in the description of inclusion and exclusion criteria also by using bullet points. This section sounds overall confusing, difficult to follow and to reproduce.
  • What about “most of Most of the PNMs were pathologically confirmed by surgical resection, or through bronchoscopic intervention. Please clarify.
  • Please describe the PET parameters by adding the specifications. Moreover, ROC curve analysis should be moved in statistical analysis section.
  • Please add a Table in which you should describe MRI specifications. The MRI protocol has already been tested or you performed it? Please clarify.
  • Please add a dedicated paragraph in which you can describe MRI and PET analysis of lung nodules. In addition, please consider rewriting pulmonary nodules analysis in a clearer manner, it seems overall confusing both on PET and MRI.
  • Please avoid the description of ROC curve analysis in MRI section.
  • Statistical analysis seems to be too reductive; please clarify how did you obtain accuracy, sens and spec of PET and MRI sequences.

Results:

  • Overall, the section seems too confusing. You should divide into several paragraphs (e.g. clinical data, roc analysis…) in which describing all results obtained in each step of analysis.
  • Please rewrite the first paragraph (L191-198), you might describe the results in a more structured manner by avoiding to use the figures to describe these.
  • You should structure the section in a clearer manner, it is difficult to follow. Each comparison made should be clarified also in MM section.

Discussion:

  • In the first paragraph you should sum up the main results obtained.
  • Please consider introducing the main limitations of PET and the adding value of MRI also in clinical setting.
  • What about the use of radiomics in differential diagnosis between benign and malignant lung nodules? 
  • MRI could be an adding tool for the clinicians?
  • Please discuss in a more structured manner your results in comparison with previous studies.
  • Your paper could have a role in clinical management of lung nodules workup? Please discuss.

References: Please see comments above and add PMID: 34072366 in discussion section.

Tables and Figures:

  • Please add a flow-chart for describing inclusion and exclusion criteria.
  • Figures 1-3 need some layout improvements, please remove the annotations.
  • Please add some Tables in which you can summarize the results obtained, also clinical data. Only two Table seems too reductive.
  • Table 2 needs some strong layout improvements.
  • Overall, please avoid the abbreviations in the captions.

Linguistic and typewriting: English writing needs some important improvements.